# PepGlider: Attribute Regularized VAE for Interpretable and Controllable Peptide Design

## Abstract

Computational peptide design requires precise control over physicochemical properties that often exhibit complex correlations. Existing generative models rely on simplistic discrete conditioning mechanisms rather than precise targeting of specific property values. We present PepGlider, a continuous attribute regularization framework that enables direct control over specific attribute values. The method achieves structured latent space and displays smooth property gradients with superior disentanglement quality. Experimental results demonstrate that PepGlider enables independent control of naturally correlated properties, and supports both unconstrained generation and targeted optimization of existing peptides. PepGlider applied to antimicrobial peptide design allows generation of candidates with desired antibacterial activity profile. Unlike existing approaches, PepGlider provides precise control over continuous property distributions while maintaining generation quality, offering a generalizable solution for therapeutic and materials applications requiring exact property specifications.

## 1 Introduction

Peptide design across diverse biomedical applications confronts a fundamental optimization challenge: achieving precise control over continuous peptide properties that often exhibit complex correlations or direct conflicts. Antimicrobial peptide (AMP) design has emerged as particularly urgent due to the escalating antimicrobial resistance crisis. Multidrug-resistant pathogens cause over 700,000 deaths annually, with projections reaching 10 million by 2050 without intervention (O'Neill, 2016). AMPs offer promising alternatives with broad-spectrum activity, rapid bacterial killing kinetics, and reduced resistance development compared to conventional antibiotics (Hancock & Sahl, 2006). The need for controllable peptide design is particularly evident in AMPs, where antimicrobial efficacy depends on complex interplay between physicochemical properties such as charge or hydrophobicity. Introducing positively charged amino acids to increase net charge - crucial for membrane interaction - often disrupts the distribution of hydrophobic residues essential for bacterial killing. Such property interdependencies complicate the optimization of antimicrobial activity, exemplifying the broader challenge of achieving precise control over functional outcomes through correlated molecular characteristics.

Deep generative models have emerged as powerful tools for peptide sequence design, but current approaches exhibit significant limitations in controllability and precision. Existing conditional generation frameworks rely predominantly on discrete conditioning mechanisms that fail to enable precise targeting of specific continuous property values. This limits optimization to coarse-grained categories rather than exact property ranges required for functional applications (Szymczak & Szczurek, 2023). Recent advances in attribute-controllable generation, particularly AR-VAE (Pati & Lerch, 2021), structure latent spaces such that specific dimensions encode target attributes through monotonic relationships. However, AR-VAE's discrete signum-based regularization of the loss function only enable relative ordering between samples, not precise targeting of specific property values. This limitation renders AR-VAE unsuitable for peptide design applications, where achieving functional outcomes requires precise control over exact property ranges.

To address these challenges in controllable peptide design, we present PepGlider, a continuous attribute regularization framework that achieves precise, independent control over correlated peptide properties. Our approach makes three main contributions: (i) we extend AR-VAE with continuous

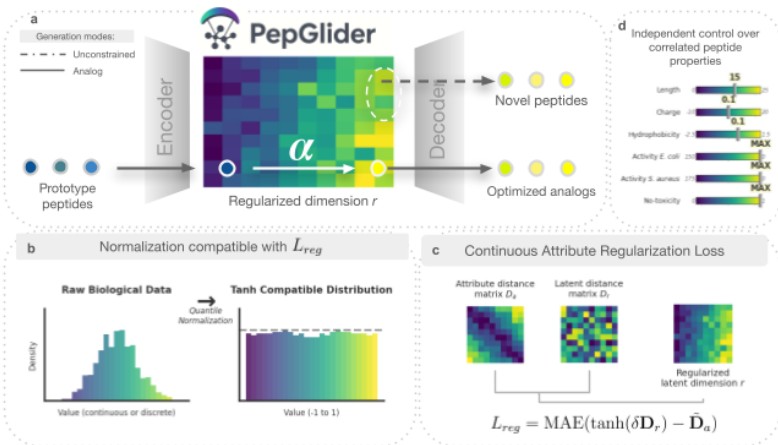

Figure 1: **PepGlider: Continuous attribute regularization for controlled peptide design.**

loss formulations that enable precise targeting of specific peptide property values rather than relative orderings, (ii) we introduce attribute-specific normalization that ensures mathematical compatibility with continuous loss while preserving biological relevance, particularly adaptive range normalization for clinically relevant antimicrobial potency ranges, and (iii) we demonstrate independent manipulation of naturally correlated physicochemical properties through structured latent space design, enabling multi-objective optimization across conflicting attributes. The resulting framework offers a generalizable methodology for precise property control across diverse peptide design applications.

## 2 METHODS

### 2.1 BACKGROUND

We build PepGlider upon the Attribute-Regularized VAE framework (Pati & Lerch, 2021), which structures latent representations such that specific dimensions encode target attributes in a monotonic fashion. Let $\mathbf{x}$ denote a data sample (here, a peptide sequence) and $\mathbf{z}$ represent the corresponding latent representation obtained through the VAE encoder. For each mini-batch of size $m$, the method constructs attribute and latent distance matrices for all pairs of samples $i, j \in \{1, ..., m\}$:

$$\mathbf{D}_a(i,j) = a(\mathbf{x}_i) - a(\mathbf{x}_j) \tag{1}$$
$$\mathbf{D}_r(i,j) = z_i^r - z_j^r \tag{2}$$

where $a(\cdot)$ represents the attribute function, $r$ denotes the regularized latent dimension, $\mathbf{D}_a$ is the attribute distance matrix, and $\mathbf{D}_r$ is the latent distance matrix. The regularization loss enforces alignment between these distance matrices:

$$\mathcal{L}_{\text{attr}} = \text{MAE}(\tanh(\delta \mathbf{D}_r) - \text{sign}(\mathbf{D}_a)) \tag{3}$$

The complete AR-VAE objective combines this with standard VAE components:

$$\mathcal{L}_{\text{AR-VAE}} = \mathcal{L}_{\text{recon}} + \beta \mathcal{L}_{\text{KL}} + \gamma \sum_{r,a} \mathcal{L}_{\text{attr}}, \tag{4}$$

where $\beta$ controls the weight of the Kullback-Leibler regularization term, while the parameter $\delta$ controls the spread of latent representations, and $\gamma$ weights the overall attribute-based regularization strength.

### 2.2 PEPGLIDER FRAMEWORK

PepGlider is designed for general peptide design applications requiring precise property control across diverse peptide optimization objectives.

### 2.2.1 Continuous Attribute Regularization

The discrete nature of the signum function in AR-VAE creates discrete comparisons that limit controllability to relative ordering rather than absolute values. We address this limitation through two key innovations:

**Continuous Attribute-Based Regularization**  We replace the signum-based comparison with a continuous regularization formulation. The modified PepGlider loss becomes:

$$\mathcal{L}_{\text{PepGlider}} = \mathcal{L}_{\text{recon}} + \beta\mathcal{L}_{\text{KL}} + \gamma\sum_{r,a}\mathcal{L}_{\text{reg}} \tag{5}$$

where the continuous property regularization term becomes:

$$\mathcal{L}_{\text{reg}} = \text{MAE}(\tanh(\delta\mathbf{D}_r) - \tilde{\mathbf{D}}_a) \tag{6}$$

**Attribute Normalization**  $\tilde{\mathbf{D}}_a$ represents the attribute distance matrix scaled to $[-1, 1]$. This formulation enables targeting of specific absolute property values rather than relative comparisons.

The continuous framework maintains gradient information throughout optimization, enabling fine-grained control while preserving numerical stability.

### 2.2.2 Controlled Peptide Generation Modes

Post-training, PepGlider enables two generation modes that leverage the structured latent space for different peptide design objectives.

**Unconstrained generation**  samples latent codes $\mathbf{z} \sim \mathcal{N}(\mathbf{0}, \mathbf{I})$ from the prior distribution and applies decoder transformations $\hat{\mathbf{x}} = \text{Dec}(\mathbf{z})$ to produce diverse peptides with desired properties that reflect the learned distribution of natural sequences. This mode enables exploration of the full peptide design space without specific property constraints from inputed peptides.

**Analog generation**  enables targeted modification of existing peptides through latent space manipulation, where a prototype sequence $\mathbf{x}$ is encoded to $\mathbf{z} = \text{Enc}(\mathbf{x})$, modified via $\alpha$ displacement $\tilde{\mathbf{z}} = \mathbf{z} + \boldsymbol{\alpha}$ to optimize specific attribute objectives, and reconstructed as $\hat{\mathbf{x}} = \text{Dec}(\tilde{\mathbf{z}})$. Here, $\boldsymbol{\alpha}$ represents the attribute shift vector that directs the latent code toward desired property values in the structured latent space.

## 3 Experimental setup

Datasets are described in Appendix A.4.

### 3.1 Attributes

**Physicochemical features**  PepGlider targets three fundamental physicochemical properties that serve as key determinants of antimicrobial activity: **net charge** (C, calculated at physiological pH), **hydrophobicity** (H, average across the sequence), and **sequence length** (L). The selection and computational implementation of these attributes is described in A.4.2.

**Antimicrobial activity against *E. coli* and *S. aureus***  We incorporate Minimum Inhibitory Concentration (MIC) values as continuous attributes predicted by APEX (Wan et al., 2024), a deep learning model for antimicrobial activity prediction trained on experimentally validated data. We obtain pathogen-specific MIC predictions for two key bacterial species: MIC against *Escherichia coli* (*E. coli*), averaged over predictions for strains ATCC 11775, AIG222, and AIG221; and MIC against *Staphylococcus aureus* (*S. aureus*), averaged over ATCC 12600 and ATCC BAA-1556 MRSA.

**Non-toxicity**  To enable safety assessment of generated peptides, we trained a binary classifier to predict hemolytic toxicity. Hemolytic activity data was extracted from the DBAASP database, focusing on HC50 measurements (peptide concentration causing 50% hemolysis). Raw toxicity values

underwent rule-based binarization (detailed procedure in A.4.3). Peptide sequences were featurized using a comprehensive set of physicochemical property values, comprizing over 100 molecular descriptors including basic properties (length, charge, hydrophobicity), structural descriptors (secondary structure fractions, topological features), and specialized amino acid scales. An XGBoost classifier was trained on these features to predict binary non-toxicity (1 = non-toxic, 0 = toxic).

**Attribute Normalization**  We employ quantile normalization (QN) for physicochemical properties and introduce adaptive range normalization for MIC values, providing granular representation of clinically relevant ranges (0-32 $\mu$g/ml) while maintaining the $[-1, 1]$ scaling required for direct alignment with $\tanh(\delta \mathbf{D}_r)$ outputs. Technical implementation details are in Appendix A.2.1, with data distribution before and after normalization procedure shown in Figure 7.

## 3.2 BASELINES

We evaluate PepGlider against six baseline approaches that represent different paradigms for controlled peptide generation and enable systematic assessment of our methodological contributions. These include ablation variants to isolate the impact of our key innovations (VAE, AR-VAE, PepGlider w/o QN, PepGlider w/ sign, PepGlider w/ z-norm), described in A.4.4, and established VAE-based models for controlled AMP generation (HydrAMP (Szymczak et al., 2023), Transformer-128 (Renaud & Mansbach, 2023)), with detailed descriptions in A.4.5.

## 3.3 EVALUATION METHODOLOGY

Implementation details and training procedure are in Appendix A.4.6. We evaluate our continuous attribute regularization framework across two complementary aspects: fundamental controllability capabilities and domain-specific application. First, we assess core framework capabilities required for controllable peptide design, including latent space disentanglement quality, continuous property control precision, and independent manipulation of correlated physicochemical properties. Second, we demonstrate framework applicability through antimicrobial peptide optimization, showcasing how general controllability enables complex, domain-specific biological objectives. Generated peptides are evaluated using antimicrobial activity predictions, safety assessment, sequence quality metrics (validity, diversity, novelty, antimicrobial potential), and disentanglement quality measures. Detailed evaluation methodology is provided in A.4.7

## 4 RESULTS

## 4.1 DISENTANGLEMENT QUALITY

Effective disentanglement is crucial for controllable generation, as it determines whether individual attributes can be manipulated independently through latent space traversal without unintended side effects on other properties at the same time. Following Pati & Lerch (2021), we assess PepGlider's disentanglement quality using five established objective metrics: Interpretability, Spearman Correlation Coefficient (SCC), Modularity, Mutual Information Gap (MIG), and Separated Attribute Predictability (SAP) averaged across charge, length, and hydrophobicity (Appendix A.4.7, Table 1).

| Model | Interpretability (↑) | SCC (↑) | Modularity (↑) | MIG (↑) | SAP (↑) |
|---|---|---|---|---|---|
| VAE | 0.175 | 0.389 | 0.833 | 0.003 | 0.023 |
| HydrAMP | 0.231 | 0.487 | **0.864** | 0.012 | 0.025 |
| Transformer-128 | 0.104 | 0.365 | 0.845 | 0.005 | 0.039 |
| AR-VAE | 0.954 | **0.995** | 0.984 | 0.450 | 0.741 |
| PepGlider w/ signum | 0.955 | **0.995** | 0.984 | 0.453 | 0.739 |
| PepGlider w/o normalization | **0.981** | **0.995** | **0.987** | **0.479** | **0.771** |
| PepGlider w z-score normalization | 0.966 | **0.995** | **0.987** | 0.478 | 0.753 |
| PepGlider | 0.931 | **0.995** | 0.985 | 0.449 | 0.719 |

Table 1: **Disentanglement quality metrics for PepGlider and baseline and ablation methods**. Mean scores across three peptide attributes (charge, length, hydrophobicity). Higher scores indicate better disentanglement for all metrics. Attribute-specific results in Table 5.

All AR-VAE variants, including PepGlider, achieve substantially superior disentanglement compared to baseline methods (VAE, HydrAMP, Transformer-128). AR-VAE and its variants demonstrate near-perfect SCC scores ($\geq 0.995$) and high performance across most metrics, with ablation variants occasionally outperforming PepGlider itself. This pattern validates that our modifications preserve the strong disentanglement properties of the original AR-VAE framework while enabling continuous control. The moderate Modularity scores for AR-VAE variants reflect biologically realistic attribute interdependencies in peptide properties, while the dramatic improvements in Interpretability (0.931 vs. 0.231 for best baseline) and MIG (0.449 vs. 0.012) demonstrate effective latent space organization for controllable generation.

### 4.1.1 CONTINUOUS ATTRIBUTE CONTROL

Practical peptide design requires smooth, predictable property transitions during latent space traversal to enable systematic peptide optimization toward target properties values. We evaluate PepGlider's continuous control capability using 2D attribute surface plots across regularized latent dimensions and an example, not regularized one. Latent vectors are systematically sampled, decoded, and evaluated for property values.

The resulting surfaces demonstrate smooth, continuous transitions across length, charge, and hydrophobicity (Figure 2, upper panel), enabling precise navigation through the latent space. The quality of latent space traversal is the highest for PepGlider when contrasted with baseline models (Appendix A.5.3). Critically, PepGlider maintains consistently high validity throughout latent space navigation, measured as FBD to training data (A.4.7). Validity assessment across systematic latent shifts (Figure 2, lower panel) shows PepGlider outperforms all baseline models, including VAE, Transformer, and HydrAMP, while AR-VAE variants perform similarly to PepGlider. PepGlider's consistent validity during traversal ensures that property optimization preserves biological plausibility of generated sequences. Additional amino acid frequency analysis confirms that PepGlider maintains realistic compositional patterns that closely match the training data distribution (Figure 9).

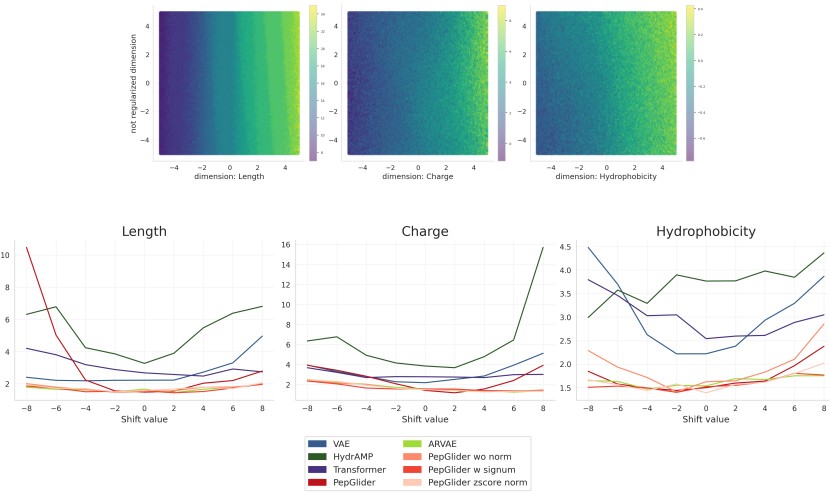

Figure 2: **Continuous attribute control through latent space manipulation. Upper panel:** PepGlider 2D attribute surface plots for length, charge, and hydrophobicity showing smooth property transitions. **Lower panel:** Validity (FBD to training data) across latent shifts for PepGlider, baselines, and ablation variants.

### 4.1.2 INDEPENDENT CONTROL OF CORRELATED PROPERTIES

The ability to manipulate correlated properties addresses conflicting optimization objectives in peptide design. We evaluate PepGlider's performance in this task through multi-attribute conditioning experiments, constraining different property combinations: individual attributes (L, C, or H), pairs (L+C, L+H, or C+H), or all three simultaneously (L+C+H), measuring target property responses while monitoring cross-interference effects.

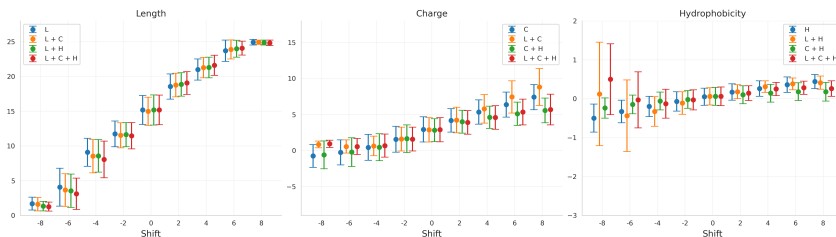

Figure 3: **Independent control of correlated peptide properties through selective attribute conditioning.** Property values across latent space shifts for **(a)** net charge, **(b)** sequence length, and **(c)** hydrophobicity under single-attribute (blue), dual-attribute (orange), and tri-attribute (green) conditioning scenarios. Error bars represent mean ± standard deviation.

PepGlider demonstrates precise independent property control across conditioning scenarios (Figure 3). Individual property constraints yield precise linear control over target attributes while leaving non-target properties unchanged. Multi-property conditioning maintains this selective control, with the exception of simultaneous charge and hydrophobicity control (C+H), which shows impairment due to inherent physicochemical constraints where hydrophobic residues are typically uncharged. Non-conflicting combinations (L+C, L+H) achieve precise multi-objective control, where each target property responds predictably to its corresponding latent dimension manipulation.

Comprehensive ablation analysis (Figure 11) reveals PepGlider achieves the greatest range of controllable values across all variants. This expanded dynamic range enables more effective targeting of specific property values and systematic exploration of property regions inaccessible to other approaches, establishing PepGlider's capability for simultaneous multi-property optimization while maintaining biological realism.

## 4.2 ANTIMICROBIAL ACTIVITY OPTIMIZATION

To demonstrate real-world applicability, we apply our framework to antimicrobial peptide optimization, where complex biological activity must be balanced against multiple physicochemical constraints. While previous sections established continuous control over basic peptide properties, practical utility depends on whether this controllability extends to biological activity predictions. Validity analysis (Figure 12) demonstrates that separating these attribute types enables more stable generation quality across the controllable space, ensuring that complex biological objectives can be pursued without compromising sequence plausibility. Therefore, we proceed with a model trained exclusively on activity and non-toxicity data.

### 4.2.1 ANTIMICROBIAL ACTIVITY CONTROL

To evaluate whether PepGlider's continuous control extends to complex biological functions, we generate 2D surface plots, where decoded peptides are evaluated using APEX MIC prediction models for *E. coli* and *S. aureus*. The smooth activity gradients across latent space demonstrate systematic control over antimicrobial potency (Figure 4, upper panels). Validation through scatterplot analysis of in-house dataset peptides projected into PepGlider's latent space reveals that experimentally verified high-activity peptides (low MIC values) naturally cluster in regions associated with predicted antimicrobial efficacy (Figure 4, lower panels), confirming that learned representations capture genuine biological function rather than arbitrary encodings.

### 4.2.2 UNCONSTRAINED GENERATION FOR HIGH-ACTIVITY PEPTIDE DISCOVERY

We evaluate PepGlider's ability to generate peptides with enhanced antimicrobial activity in the unconstrained generation mode via sampling from high-activity latent regions (see 2.2.2), by comparing its performance against established generative approaches. Two complementary assessments demonstrate both controllability and biological relevance: (1) antimicrobial potential approximated through FBD to active peptides during targeted sampling from high-activity latent regions (Figure 5, upper panel), and (2) direct APEX-predicted MIC distributions for sequences from unconstrained

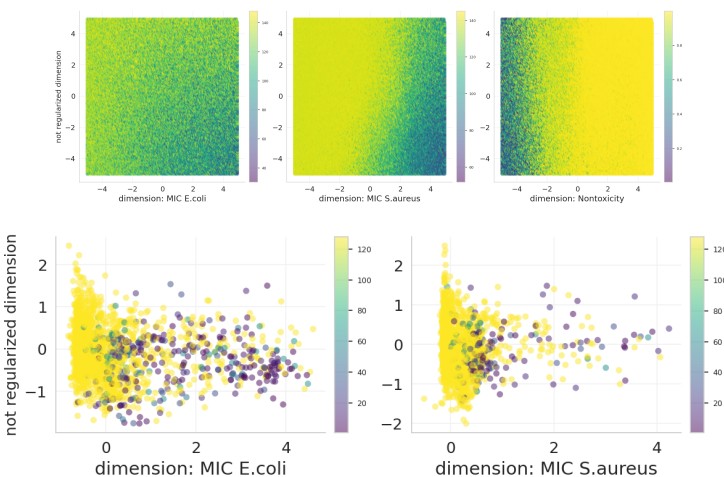

Figure 4: **Continuous antimicrobial activity control in PepGlider latent space. Upper panels:** 2D surface plots showing APEX-predicted MIC values across latent space for *E. coli* (left). *S. aureus* (middle), and non-toxicity predictions (right). **Lower panels:** Validation scatterplots showing peptides from proprietary dataset A.4.7 projected into latent space, colored by experimental MIC values for *E. coli* (left) and *S. aureus* (right).

generation (Figure 5, lower panel). Based on Figure 5, we select the best shift combination per strain (+8.0 for *E. coli*, +2.0 for *S. aureus*) and use this setup to calculate sequence metrics, including validity, novelty, and diversity, provided in Table 2.

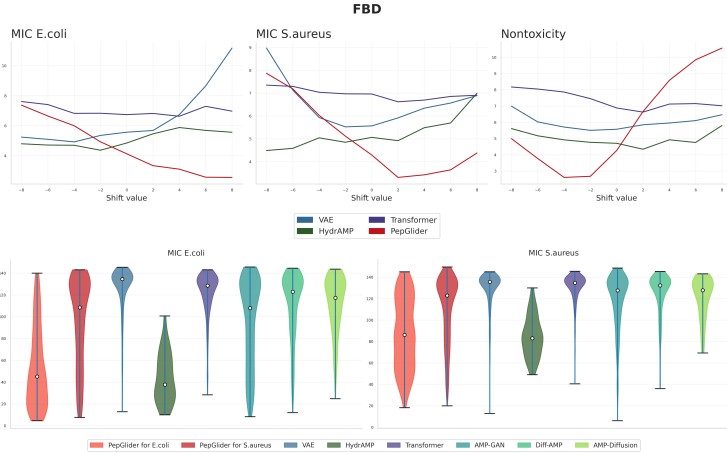

Figure 5: **High-activity peptide generation through strategic latent space sampling.** Fréchet Biological Distance (FBD) scores comparing PepGlider to baseline methods when generating peptides from latent space regions corresponding to low MIC predictions. FBD computed between generated samples and reference set of highly active antimicrobial peptides using fine-tuned ESM2 embeddings. Lower FBD scores indicate greater similarity to genuine high-activity antimicrobial peptides.

The results demonstrate PepGlider's superior performance in unconstrained generation of high-activity antimicrobial candidates. APEX predictions (Figure 5, lower panel) show that PepGlider-generated sequences achieve substantially lower MIC distributions for both *E. coli* and *S. aureus* compared to all baseline and external methods. For *E. coli*, PepGlider's unconstrained generation produces a concentrated distribution around 40-60 μg/ml with significant density below 32 μg/ml (clinically relevant threshold), while competing methods show broader distributions centered at higher MIC values. Similarly, for *S. aureus*, PepGlider maintains low MIC predictions with

| Model | Validity (↓) | AMP potential (↓) | Novelty (↑) | Diversity (↑) |
|---|---|---|---|---|
| VAE | 2.201 | 5.671 | 1.0 | 0.988 |
| HydrAMP | 3.616 | 4.950 | 1.0 | 0.897 |
| Transformer | 2.668 | 6.827 | 1.0 | **1.144** |
| AMP-GAN | 2.202 | 5.705 | 1.0 | 0.990 |
| Diff-AMP | 3.237 | 4.254 | 1.0 | 0.940 |
| $AMP_{Diffusion}$ | 3.936 | 8.036 | 1.0 | 0.825 |
| PepGlider* | **1.498** | **2.935** | 1.0 | 0.956 |

Table 2: **Sequence quality metrics for unconstrained generation across generative models.** Comparison of validity, AMP potential, novelty, and diversity for PepGlider and baseline/external generative methods. *PepGlider results averaged over *E. coli* and *S. aureus* predictions.

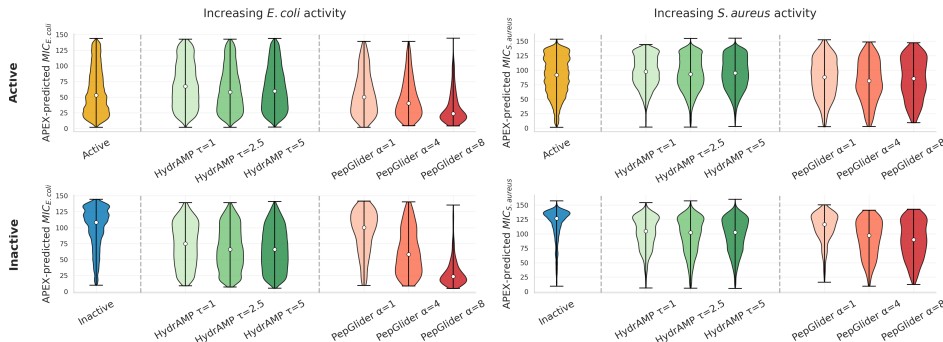

Figure 6: **Existing peptide improvement through analog generation.** Violin plots showing antimicrobial activity improvement distributions for PepGlider and HydrAMP analog generation methods across positive prototypes (initially high activity) and negative prototypes (initially low activity), measured using the APEX regressor.

dense concentration in the high-activity range, contrasting with the elevated MIC distributions of VAE, Transformer, and external generative models including AMP-GAN (Van Oort et al., 2021), Diff-AMP (Wang et al., 2024), and AMP-Diffusion (Torres et al., 2025). HydrAMP shows moderate performance, but fails to achieve PepGlider's consistent low-MIC generation. These results demonstrate that PepGlider's continuous attribute regularization framework successfully achieves practically relevant biological activity enhancement through unconstrained generation.

### 4.2.3 ANALOG GENERATION FOR OPTIMIZING AMP ACTIVITY

We evaluate PepGlider's ability to optimize existing peptides through targeted latent manipulation, and we compare against HydrAMP, the only baseline model equipped with analog generation. Starting from active and inactive prototypes, we apply analog generation to enhance predicted activity against *E. coli* and *S. aureus*. HydrAMP parameters include temperature values ($\tau$ = 1, 2.5, 5) with conditioning parameters targeting high activity. PepGlider uses shift magnitudes ($\alpha$ = 1, 4, 8) controlling latent space displacement toward higher predicted activity.

PepGlider demonstrates superior activity enhancement across both bacterial targets (Figure 6). Starting from both active and inactive prototypes, PepGlider consistently achieves lower predicted MIC values than HydrAMP across all parameters, with $\alpha$=8 reaching clinically relevant potency around 32 µg/mL. PepGlider shows robust improvement even from inactive peptides, effectively transforming low-activity sequences into promising candidates. *S. aureus* activity improvement proves more challenging than *E. coli*, particularly from already active prototypes. Notably, PepGlider enables independent optimization for each bacterial strain through separate regularized dimensions, a capability rarely achieved in antimicrobial peptide design frameworks where strain-specific targeting typically requires distinct models or post-hoc filtering approaches Szymczak & Szczurek (2023).

### 4.2.4 MULTI-OBJECTIVE OPTIMIZATION: ACTIVITY-SAFETY TRADE-OFFS

A critical challenge in antimicrobial peptide development is balancing efficacy against safety, as enhanced activity often correlates with increased toxicity. We evaluate PepGlider's ability to navigate

this trade-off through simultaneous manipulation of MIC and non-toxicity regularized dimensions. While manipulating MIC-regularized dimensions to enhance antimicrobial activity against *E. coli* and *S. aureus*, we simultaneously apply different non-toxicity regularization strategies ($\alpha = -2, 0, +2$ for non-toxicity regularized dimension, as well as random control) to assess whether toxicity increases can be mitigated.

The results (Figure 13) demonstrate PepGlider's capacity for controlled multi-objective optimization. As expected, shifting toward lower MIC values in the regularized dimensions successfully enhances predicted antimicrobial activity for both bacterial targets. Critically, simultaneous non-toxicity regularization ($\alpha = +2$) helps maintain higher non-toxicity scores compared to unregularized approaches, demonstrating that PepGlider can partially decouple the activity-safety trade-off. This capability enables rational optimization of therapeutic windows, allowing researchers to enhance antimicrobial potency while minimizing safety risks.

## 5 DISCUSSION

PepGlider addresses fundamental limitations in controllable peptide design through continuous attribute regularization and adaptive normalization strategies. Our framework enables independent manipulation of correlated properties while maintaining biological plausibility, demonstrated through superior latent quality evaluation and systematic property control across challenging scenarios like activity-safety trade-offs.

Key limitations include reduced performance for inherently conflicting objectives, particularly the activity-safety trade-off where enhanced antimicrobial potency often correlates with increased toxicity. Future developments should prioritize methods that efficiently utilize sparse biological experimental data directly, reducing dependence on intermediate prediction models while maintaining controllability. The current attribute set, while comprehensive for basic physicochemical properties, could be expanded to include synthesizability constraints, structural features (secondary structure propensity, flexibility), and manufacturing considerations critical for therapeutic translation.

The continuous attribute regularization framework's versatility extends beyond antimicrobial peptides to diverse therapeutic applications, providing a flexible framework for controllable generation.

## 6 ETHICS STATEMENT

This research involves computational design of antimicrobial peptides using machine learning methods. All datasets used for training and evaluation consist of publicly available peptide sequences and experimental measurements from established databases (AMPScanner, dbAMP, DRAMP, DBAASP). No human subjects, animal experiments, or clinical trials were involved in this computational study. The potential therapeutic applications of designed antimicrobial peptides could contribute to addressing antimicrobial resistance, a significant global health challenge. However, any peptides generated by this framework require extensive experimental validation, safety testing, and regulatory approval before clinical consideration. The hemolytic toxicity predictions used in this work are computational estimates and cannot replace experimental safety assessment.

## 7 REPRODUCIBILITY STATEMENT

We provide comprehensive implementation details to ensure reproducibility. Model architectures, hyperparameters, and training procedures are detailed in the main text and appendix (Table 3). All normalization procedures, including quantile transformation and adaptive range normalization, are mathematically specified with explicit equations. Evaluation metrics and baseline comparisons use established methods with clear mathematical definitions. The proprietary validation dataset contains experimental MIC measurements that enable independent performance assessment, though specific data cannot be shared due to proprietary restrictions. Code and trained models will be made available upon publication to facilitate reproduction and extension of this work. The framework's implementation using standard deep learning libraries ensures compatibility with common research environments.

## 8 LLM USAGE

Large language models were used as writing assistance tools during the preparation of this manuscript. Specifically, Claude was employed for improving clarity and flow of technical explanations, and generating alternative phrasings for complex methodological concepts, proofreading and copy-editing assistance.

All scientific content, experimental design, results, and conclusions are entirely the work of the human authors. LLMs were not used for experimental design decisions, or scientific reasoning. The core contributions, methodological innovations, and technical implementations represent original research by the authors. All factual claims and experimental results were verified independently by the research team.

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

# A  APPENDIX

## A.1  RELATED WORK

Controllable peptide design intersects multiple research areas, including conditional generation and latent space regularization, each addressing different aspects of the challenge of navigating correlated peptide properties.

**Controllable Peptide Design**  Current approaches to controllable peptide generation, particularly for AMPs, employ three main strategies: conditional generation, post-hoc filtering, and guidance during sampling. While conditional methods like HydrAMP (Szymczak et al., 2023) directly incorporate constraints, they are limited to binary classification or struggle with multiple objectives. Post-hoc approaches (Das et al., 2021; Pandi et al., 2023; Torres et al., 2025) suffer from severe efficiency limitations when targeting rare property combinations. The exponential search space of peptide sequences makes exhaustive sampling impractical, particularly when multiple properties must be optimized simultaneously. Guidance-based methods attempt to steer the generation process toward desired properties during sampling, including approaches that use Monte Carlo Tree Guidance (Tang et al., 2025) and reinforcement learning with property-based rewards (Wang et al., 2024). However, guidance approaches operate at the sampling level rather than embedding controllability into the learned representation, making them computationally expensive during generation and unable to leverage the structured relationships between properties for more efficient optimization.

**Latent Space Regularization**  Latent space regularization methods from other domains learn representations where properties naturally align with latent structure. VAE-based approaches have pioneered this direction through various regularization strategies. CorrVAE (Wang et al., 2022) addresses property correlations through specialized loss functions designed to handle interdependent data characteristics. Property-controllable VAE (Guo et al., 2020) incorporates property prediction losses directly into the variational objective, creating latent representations that encode desired features. Conditional Subspace VAE (Klys et al., 2018) partitions the latent space according to property-specific regions, enabling targeted sampling from relevant subspaces. AR-VAE (Pati & Lerch, 2021) aligns latent and attribute spaces through distance matrix matching to create monotonic relationships between latent dimensions and target properties. While these latent space methods offer promising frameworks for controllable generation, they have not been adapted to address the specific challenges of peptide design, particularly the need for precise property targeting across correlated physicochemical characteristics and efficient access to rare, but functional attribute combinations essential for therapeutic applications.

## A.2  EXTENDED METHODS

### A.2.1  ATTRIBUTE NORMALIZATION PROCEDURES

We introduce attribute-specific normalization strategies as a core methodological contribution that ensures compatibility with our continuous loss formulation while preserving biological meaning.

**Quantile Normalization** Applied to charge, length, hydrophobicity, and non-toxicity predictions. Raw values are transformed via quantile transformation $Q(\cdot)$ to uniform distribution $U(0, 1)$, then linearly scaled:

$$\tilde{p}_i = 2Q(p_i) - 1 \tag{7}$$

This ensures uniform property space coverage and eliminates scale bias while maintaining the required $[-1, 1]$ range.

**Adaptive Range Normalization** A normalization strategy that addresses the clinical importance of low MIC values while maintaining loss compatibility. The approach allocates 70% of the normalized range to therapeutically relevant concentrations (0-32 $\mu$g/ml) and 30% to higher values:

*Higher concentrations (>32 $\mu$g/ml) $\rightarrow$ [-1, -0.4]:*

$$\tilde{p}_{out} = -1 + 0.6 \cdot \text{CDF}_{out}(p) \tag{8}$$

*Clinically relevant range (0-32 $\mu$g/ml) $\rightarrow$ [-0.4, 1]:*

$$\tilde{p}_{ROI} = -0.4 + 1.4 \cdot \text{CDF}_{ROI}(p) \tag{9}$$

where $\text{CDF}_{ROI}$ and $\text{CDF}_{out}$ are empirical cumulative distribution functions computed within each region using histogram-based quantile mapping.

This normalization framework enables precise property control by ensuring all normalized attributes operate within the same bounded range as the latent regularization terms. Unlike the discrete approach that rely on signum function, our continuous formulation maintains gradient information throughout optimization, enabling fine-grained control over peptide properties while preserving compatibility with standard VAE training procedures.

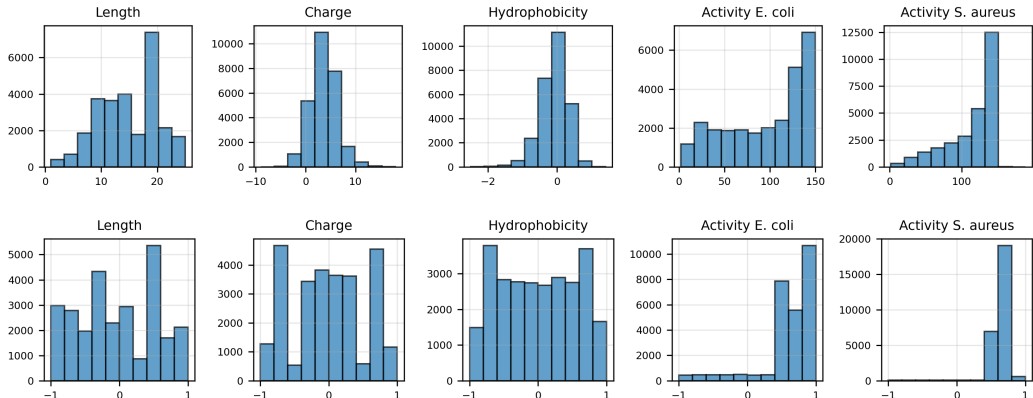

Figure 7: **Two-stage normalization procedure for continuous attribute regularization.** Distribution comparison of peptide properties before (upper panel) and after (lower panel) the normalization procedure.

## A.3    EXTENDED EXPERIMENTAL SETUP

## A.4    DATASETS

The training dataset comprises 27331 curated antimicrobial peptide sequences derived from four comprehensive databases: AMPScanner (Veltri et al., 2018), dbAMP (Yao et al., 2025), DRAMP (Kang et al., 2019), and DBAASP (Pirtskhalava et al., 2021). Sequences are restricted to a maximum of 25 amino acid residues, capturing the predominant length range of naturally occurring antimicrobial peptides. Duplicate sequences are removed across databases to ensure unique representation within the training corpus.

### A.4.1    EVALUATION DATASET

For evaluation of PepGlider, we utilize a proprietary dataset containing experimental MIC measurements for 1,736 peptides tested against 11 clinically relevant bacterial strains. The dataset includes measurements against Gram-negative bacteria (*A. baumannii* ATCC 19606, *E. coli* ATCC 11775, *E. coli* AIC221, carbapenem-resistant *E. coli* AIC222, *K. pneumoniae* ATCC 13883, *P. aeruginosa* PAO1 and PA14) and Gram-positive bacteria (*S. aureus* ATCC 12600, methicillin-resistant *S. aureus* ATCC BAA-1556, vancomycin-resistant *E. faecalis* ATCC 700802, and vancomycin-resistant *E. faecium* ATCC 700221). This comprehensive dataset enables validation of generated peptides against both standard reference strains and clinically significant drug-resistant isolates, providing robust assessment of antimicrobial activity across diverse bacterial targets.

### A.4.2    PHYSICOCHEMICAL PROPERTIES

We selected net charge, hydrophobicity, and sequence length as target attributes for PepGlider based on their established roles in antimicrobial peptide function and their ability to discriminate between active and inactive peptides.

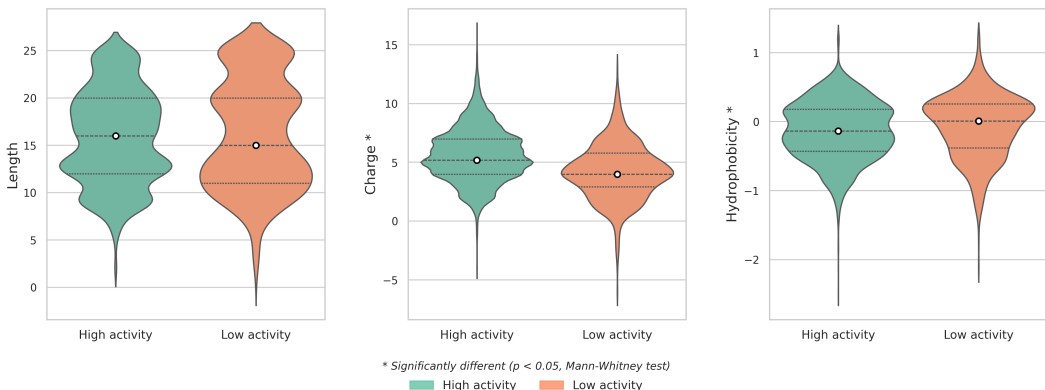

Figure 8: **Discriminative physicochemical properties between active and inactive antimicrobial peptides**. Distribution comparison of net charge, hydrophobicity, and sequence length between active peptides (green) and inactive peptides (orange) from the curated dataset.

**Net Charge**   plays a critical role in the initial electrostatic interactions between cationic antimicrobial peptides and negatively charged bacterial membranes (Yeaman & Yount, 2003). Positively charged residues facilitate binding to bacterial lipopolysaccharides and phospholipids, while excessive charge can lead to reduced membrane permeation and potential cytotoxicity (Hancock & Sahl, 2006). Net charge is calculated at physiological pH (7.0), accounting for the ionization state of terminal groups and ionizable side chains based on their respective pKa values from the *modlamp* implementation.

**Hydrophobicity**   determines the peptide's ability to partition into and disrupt bacterial membranes (Wieprecht et al., 1997). Optimal hydrophobic content enables membrane insertion while preventing aggregation or excessive hemolytic activity. Average hydrophobicity is computed using established amino acid hydrophobicity scales:

$$H(\mathbf{x}) = \frac{1}{L} \sum_{i=1}^{L} h_i \tag{10}$$

where $h_i$ represents the hydrophobicity value for amino acid $i$ and $L$ is the sequence length.

**Sequence length**   constrains both structural flexibility and membrane interaction mechanisms. Shorter peptides typically adopt extended conformations that facilitate membrane carpet formation, while longer sequences may form more complex secondary structures affecting activity and selectivity (Shai & Oren, 2001). Sequence length is a direct enumeration of amino acid residues:

$$L(\mathbf{x}) = |\mathbf{x}| \tag{11}$$

Physicochemical properties are computed using the *modlamp* (Müller et al., 2017) package, implementing algorithms for antimicrobial peptide analysis:

To validate these properties as discriminative features, we analyzed their distributions across active (MIC $\leq$ 32 $\mu$g/ml) and inactive (MIC $\geq$ 128 $\mu$g/ml) peptides in our curated dataset (Figure 8). Active peptides exhibit distinct distributions for all three properties: moderate positive charge (mean ± SD), intermediate hydrophobicity values, and concentrated length distributions around 10-25 residues. These clear distributional differences support their selection as target attributes for continuous control in the PepGlider framework.

### A.4.3   Non-toxicity

**Data Extraction and Preprocessing**   Hemolytic activity measurements were extracted from DBAASP (Pirtskhalava et al., 2021), focusing on records containing HC50 values and percentage

hemolysis data. Activity measure values and groups were normalized to lowercase for consistent processing. Percentage hemolysis values were extracted using regex parsing with two prioritized patterns: (1) values with standard deviations (e.g., "15.2±3.1% hemolysis"), taking the primary value before ±, and (2) range formats (e.g., "10-20% hemolysis"), using the midpoint average.

**Binary Toxicity Classification Rules**    For peptides with single measurements, the following hierarchy was applied:

- Direct non-toxic assignment: 0% hemolysis records → non-toxic (1)
- Activity-based toxic assignment: activity $\leq$32 $\mu$g/mL AND >1% hemolysis → toxic (0)
- Activity-based non-toxic assignment: activity >32 $\mu$g/mL AND $\leq$10% hemolysis → non-toxic (1)
- Hemolysis threshold: >50% hemolysis → toxic (0)
- HC50-specific rules: HC50 $\leq$256 $\mu$g/mL → toxic (0), otherwise non-toxic (1)

For peptides with multiple measurements, consensus rules were applied:

- Any measurement <32 $\mu$g/mL → toxic (0)
- All measurements >128 $\mu$g/mL → non-toxic (1)
- All measurements $\leq$10% hemolysis → evaluate based on activity (<32 $\mu$g/mL threshold for toxic (0))
- HC50 measurements prioritized when available, using 128 $\mu$g/mL threshold (toxic (0) if $\leq$128 $\mu$g/mL)

Peptides not meeting any classification criteria were excluded from the training dataset.

**Feature Engineering**    The physicochemical property calculation framework computed 100+ features per sequence, including:

- Basic properties: length, charge, isoelectric point, molecular weight, aromaticity
- Hydrophobicity scales: AASI, Argos, Eisenberg, GRAVY, Kyte-Doolittle (16 scales total)
- Structural descriptors: secondary structure fractions, flexibility, entropy
- Specialized scales: Z-scales (5D), Kidera factors (10D), VHSE scales (8D), FASGAI vectors (6D)
- Topological features: polar surface area, H-bond acceptors/donors, rotatable bonds
- Compositional features: amino acid frequencies, structural class distributions

**Model Training and Validation**    XGBoost classifier hyperparameters were optimized on the training set with 974,582 peptides (1,157 toxic and 973,425 non-toxic) balanced using focal loss set to handle the imbalanced dataset. Model performance was assessed on the independent HydrAMP dataset containing experimentally validated antimicrobial peptides with known hemolytic profiles (Accuracy = 0.8333, F1-Score = 0.9048).

A.4.4    ABLATIONS

**VAE Baseline** serves as our primary ablation control, employing the identical transformer-based VAE architecture as PepGlider with Importance Weighted Autoencoder components and $\beta$-VAE regularization. The model is trained on the same dataset with identical quantile normalization, but without the continuous attribute regularization loss ($\gamma = 0$) and and 10-times decreased $\beta$. This configuration isolates the contribution of our continuous regularization framework from the architectural and preprocessing components.

**AR-VAE** (Pati & Lerch, 2021) represents the original attribute regularization formulation using the signum function for discrete ordinal comparisons and standard preprocessing without quantile normalization. This baseline evaluates the impact of our continuous loss formulation and normalization improvements.

**PepGlider w/o QN** removes quantile normalization while retaining the continuous loss formulation and signum replacement. Raw attribute values are used directly in the loss computation, isolating the contribution of the normalization procedure.

**PepGlider w/ sign** retains the original signum function from AR-VAE while incorporating our quantile normalization approach. This variant evaluates whether normalization alone can improve discrete attribute regularization.

**PepGlider w/ z-norm** replaces quantile normalization with standard z-score normalization (zero mean, unit variance), testing alternative normalization strategies while maintaining the continuous loss formulation.

### A.4.5 BASELINE MODELS

**HydrAMP** (Szymczak et al., 2023) is a conditional variational autoencoder for antimicrobial peptide generation. The model employs Jacobian-based disentanglement regularization to enforce independence between latent representations $\mathbf{z}$ and discrete conditioning variables ($c_{AMP}, c_{MIC} \in \{0, 1\}$). Property control is achieved through conditional decoding $\text{Dec}(\mathbf{z}, c)$ with binary labels for antimicrobial activity and potency. In contrast to PepGlider's continuous attribute regularization in latent space, HydrAMP guides generation through discrete conditions fed directly to the decoder. The model supports unconstrained generation and temperature-controlled analogue generation modes.

**Transformer-128** (Renaud & Mansbach, 2023) employs a transformer-based autoencoder architecture with a 128-dimensional latent space for peptide generation. The model learns implicit partitioning of the latent space into regions corresponding to high and low AMP probabilities without explicit incorporation of mechanisms for continuous property control.

### A.4.6 PEPGLIDER IMPLEMENTATION

PepGlider employs a transformer-based VAE architecture (Kingma & Welling, 2013) optimized for variable-length biological sequences. The encoder $\text{Enc}(\cdot)$ maps peptide sequences $\mathbf{x} \in \{A, C, D, ..., Y\}^L$ to latent representations $\mathbf{z} \in \mathbb{R}^d$ through CLS token aggregation, where a learnable classification token attends to all sequence positions via multi-head self-attention mechanisms. The encoder outputs parameterize a Gaussian posterior $q(\mathbf{z}|\mathbf{x})$ with mean $\boldsymbol{\mu}(\mathbf{x})$ and standard deviation $\boldsymbol{\sigma}(\mathbf{x})$. The decoder $\text{Dec}(\cdot)$ reconstructs sequences from latent codes by replicating the latent vector across sequence positions and applying positional encodings for position-specific token generation. The architecture incorporates $\beta$-VAE regularization (Higgins et al., 2017) and Importance Weighted Autoencoder components (Burda et al., 2015) to enhance posterior distribution approximation and latent space disentanglement capabilities.

All models were trained on NVIDIA A100 GPUs with 8GB memory. PepGlider and baseline models were trained for approximately 48 hours. The hyperparameter details are presented in Table 3.

**Training Schedule:** The $\beta$ and $\gamma$ parameter follow linear annealing schedules from their initial to final values over the specified warmup steps, after which they remain constant. The KL divergence weight $\beta$ gradually increases to prevent posterior collapse.

### A.4.7 EVALUATION METHODOLOGY

**Antimicrobial Activity and Non-toxicity Assessment** We evaluate antimicrobial activity using APEX (Wan et al., 2024) predictions for *E. coli* and *S. aureus* providing species-specific MIC predictions for clinically relevant pathogens. Toxicity assessment is performed using our trained hemolytic toxicity classifier as described in A.4.3

**Fréchet Biological Distance** To evaluate the quality of generated peptides in biologically relevant embedding space, we compute Fréchet Biological Distance (FBD) using fine-tuned ESM2 embeddings. The ESM2-t12 model (Lin et al., 2023) was fine-tuned for binary antimicrobial activity classification using active/inactive labels with thresholds of $\leq 32$ $\mu$g/ml for active and $\geq 128$ $\mu$g/ml for inactive peptides.

r

Table 3: Model hyperparameters and training details.

| Parameter | Value |
|---|---|
| **Architecture** | |
| Attention Heads | 4 |
| Transformer Layers | 6 |
| Latent Dimension | 56 |
| Positional Encoding | Additive |
| Dropout Rate | 0.1 |
| Layer Normalization | Enabled |
| **Training** | |
| Optimizer | Adam |
| Learning Rate | 0.001 |
| Batch Size | 512 |
| Epochs | 3100 |
| IWAE Samples ($K$) | 10 |
| **VAE Regularization** | |
| $\beta$ Initial | 0.00001 |
| $\beta$ Final | 0.1 |
| $\beta$ Warmup Steps | 8000 |
| **Attribute Regularization** | |
| Regularized Dimensions | [0, 1, 2, 3, 4, 5] |
| $\gamma$ Initial | 0.00001 |
| $\gamma$ Final | 20 |
| $\gamma$ Warmup Steps | 8000 |
| $\gamma$ Triggered epoch | 1000 |
| $\delta$ | 0.1 for PepGlider for physicochemical attributes |
| | 0.6 for PepGlider only antimicrobial attributes |
| **Data** | |
| Max Sequence Length | 25 |
| Vocabulary Size | 20 |
| Property Normalization | Quantile (10 bins) |
| Properties Regularized | Length, Charge, Hydrophobicity, MIC E.coli, MIC S.aureus, Non-toxicity |

FBD is computed analogously to Fréchet Inception Distance (Heusel et al., 2017) by modeling the distributions of ESM2 embeddings as multivariate Gaussians:

$$\text{FBD} = ||\boldsymbol{\mu}_{\text{real}} - \boldsymbol{\mu}_{\text{gen}}||_2^2 + \text{Tr}(\boldsymbol{\Sigma}_{\text{real}} + \boldsymbol{\Sigma}_{\text{gen}} - 2(\boldsymbol{\Sigma}_{\text{real}}\boldsymbol{\Sigma}_{\text{gen}})^{1/2}) \tag{12}$$

where $\boldsymbol{\mu}$ and $\boldsymbol{\Sigma}$ represent the mean and covariance of the embedding distributions for real and generated peptides, respectively.

**Sequence Quality Metrics**

- **Validity** measures how well generated peptides conform to the training distribution by computing FBD between generated sequences and the training dataset using ESM2 embeddings.

- **Diversity** is a fraction of unique generated sequences not present in the training dataset.

- **Novelty** is cmputed using average pairwise Levenshtein distance among generated sequences.

- **AMP Potential** evaluates similarity to experimentally validated antimicrobial peptides by computing FBD to a curated set of active peptides with documented MIC $\leq 32$ $\mu$g/ml against at least one bacterial strain in DBAASP.

**Disentanglement Quality Metrics** Following Pati & Lerch (2021), we assess PepGlider's disentanglement quality using five established objective metrics:

- **Interpretability** measures how well individual latent dimensions align with specific attributes by evaluating the variance explained by the most informative dimension for each attribute:

$$\text{Interpretability} = \frac{1}{K} \sum_{k=1}^{K} \max_j R^2(a_k, z_j) \tag{13}$$

  where $K$ is the number of attributes, $a_k$ is the $k$-th attribute, and $R^2(a_k, z_j)$ is the coefficient of determination between attribute $k$ and latent dimension $j$.

- **Spearman Correlation Coefficient (SCC)** quantifies the monotonic relationship between latent dimensions and target attributes:

$$\text{SCC} = \frac{1}{K} \sum_{k=1}^{K} \max_j |\rho_s(a_k, z_j)| \tag{14}$$

  where $\rho_s$ denotes the Spearman correlation coefficient.

- **Modularity** assesses whether each attribute is controlled by a distinct set of latent dimensions, measuring the degree of separation between attribute-dimension associations.

- **Mutual Information Gap (MIG)** evaluates disentanglement by measuring the difference between the top two mutual information values:

$$\text{MIG} = \frac{1}{K} \sum_{k=1}^{K} \frac{I(a_k; z_{j^{(1)}}) - I(a_k; z_{j^{(2)}})}{H(a_k)} \tag{15}$$

  where $j^{(1)}$ and $j^{(2)}$ are the indices of the latent dimensions with highest and second-highest mutual information with attribute $k$, and $H(a_k)$ is the entropy of attribute $k$.

- **Separated Attribute Predictability (SAP)** measures how well attributes can be predicted from individual latent dimensions while being unpredictable from others, indicating effective separation of attribute control.

Results across all metrics are reported in Table A.5.1.

## A.5 EXTENDED RESULTS

### A.5.1 DISENTANGLEMENT DETAILED ANALYSIS

| Model | Attribute | Interpretability (↑) | SCC (↑) | Modularity (↑) | MIG (↑) | SAP (↑) |
|---|---|---|---|---|---|---|
| VAE | Length | 0.542 | 0.754 | 0.893 | 0.012 | 0.022 |
| | Charge | 0.088 | 0.320 | 0.660 | 0.001 | 0.026 |
| | Hydrophobicity | 0.080 | 0.309 | 0.736 | 0.000 | 0.009 |
| HydrAMP | Length | 0.951 | 0.973 | 0.957 | 0.056 | 0.020 |
| | Charge | 0.024 | 0.439 | 0.868 | 0.003 | 0.042 |
| | Hydrophobicity | 0.178 | 0.440 | 0.867 | 0.000 | 0.029 |
| Transformer | Length | 0.014 | 0.289 | 0.799 | 0.005 | 0.038 |
| | Charge | 0.022 | 0.333 | 0.723 | 0.004 | 0.013 |
| | Hydrophobicity | 0.261 | 0.494 | 0.975 | 0.008 | 0.121 |
| ARVAE | Length | 0.994 | 0.995 | **0.999** | 0.895 | 0.948 |
| | Charge | 0.947 | 0.994 | 0.979 | 0.280 | 0.655 |
| | Hydrophobicity | 0.921 | 0.996 | 0.177 | 0.211 | 0.621 |
| PepGlider | Length | 0.977 | 0.995 | **0.999** | 0.856 | 0.927 |
| | Charge | 0.934 | 0.991 | 0.980 | 0.292 | 0.649 |
| | Hydrophobicity | 0.881 | **0.998** | 0.976 | 0.211 | 0.587 |
| PepGlider w/o normalization | Length | 0.994 | 0.995 | **0.999** | 0.884 | 0.953 |
| | Charge | 0.958 | 0.992 | 0.981 | 0.297 | 0.670 |
| | Hydrophobicity | 0.990 | 0.997 | 0.983 | 0.257 | 0.690 |
| PepGlider with signum | Length | 0.994 | 0.995 | **0.999** | **0.896** | 0.951 |
| | Charge | 0.950 | 0.994 | 0.980 | 0.285 | 0.655 |
| | Hydrophobicity | 0.921 | 0.996 | 0.973 | 0.177 | 0.612 |
| PepGlider w z-score normalization | Length | **0.996** | 0.995 | **0.999** | 0.882 | **0.959** |
| | Charge | 0.966 | 0.982 | 0.982 | **0.321** | 0.668 |
| | Hydrophobicity | 0.937 | **0.998** | 0.980 | 0.231 | 0.631 |

Table 4: **Detailed disentanglement quality metrics for PepGlider and baseline and ablation methods**. Higher scores indicate better disentanglement for all metrics.

| Model | Attribute | Interpretability (↑) | SCC (↑) | Modularity (↑) | MIG (↑) | SAP (↑) |
|---|---|---|---|---|---|---|
| VAE | MIC *E. coli* | 0.087 | 0.287 | 0.815 | 0.001 | 0.029 |
| | MIC *S. aureus* | 0.076 | 0.275 | 0.795 | 0.001 | 0.027 |
| | Nontoxicity | 0.054 | 0.550 | 0.897 | 0.002 | 0.020 |
| HydrAMP | MIC *E. coli* | 0.000 | 0.295 | 0.928 | 0.000 | 0.020 |
| | MIC *S. aureus* | 0.000 | 0.248 | 0.931 | 0.001 | 0.012 |
| | Nontoxicity | 0.007 | 0.320 | 0.931 | 0.002 | 0.009 |
| Transformer | MIC *E. coli* | 0.129 | 0.365 | 0.945 | 0.001 | 0.008 |
| | MIC *S. aureus* | 0.090 | 0.304 | 0.932 | 0.000 | 0.006 |
| | Nontoxicity | 0.039 | 0.371 | 0.945 | 0.001 | 0.027 |
| PepGlider | MIC *E. coli* | 0.566 | 0.962 | 0.992 | 0.074 | 0.022 |
| | MIC *S. aureus* | **0.758** | 0.933 | 0.991 | 0.062 | **0.236** |
| | Nontoxicity | 0.332 | **0.992** | **0.994** | **0.075** | 0.123 |

Table 5: **Detailed attribute-specific disentanglement quality metrics results across antimicrobial properties (MIC *E. coli*, MIC *S. aureus*, nontoxicity) for PepGlider and baseline methods**. Higher scores indicate better disentanglement for all metrics.

### A.5.2 RECONSTRUCTION

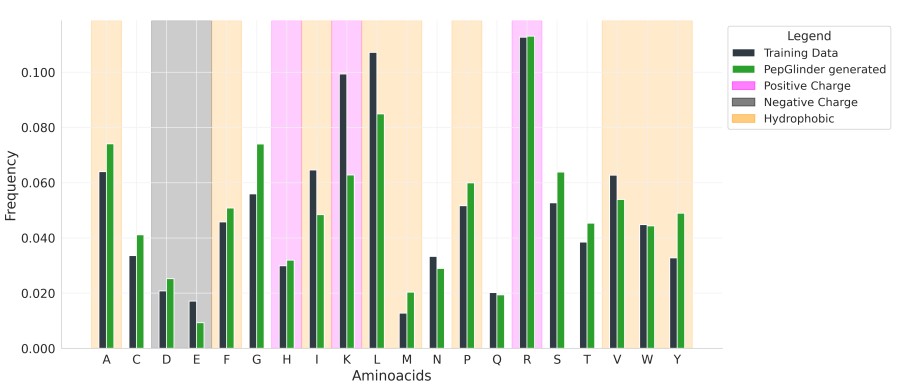

Figure 9: **Reconstruction quality assessment for PepGlider-generated peptides.** Amino acid frequency distributions comparing generated peptides (green bars) with training data (black bars), demonstrating that PepGlider maintains realistic compositional patterns despite attribute regularization constraints. Amino acids crucial for hydrophobicity are highlighted in orange, and amino acids contributing to positive charge and negative charge are highlighted in pink and gray, respectively.

### A.5.3 ABLATIONS

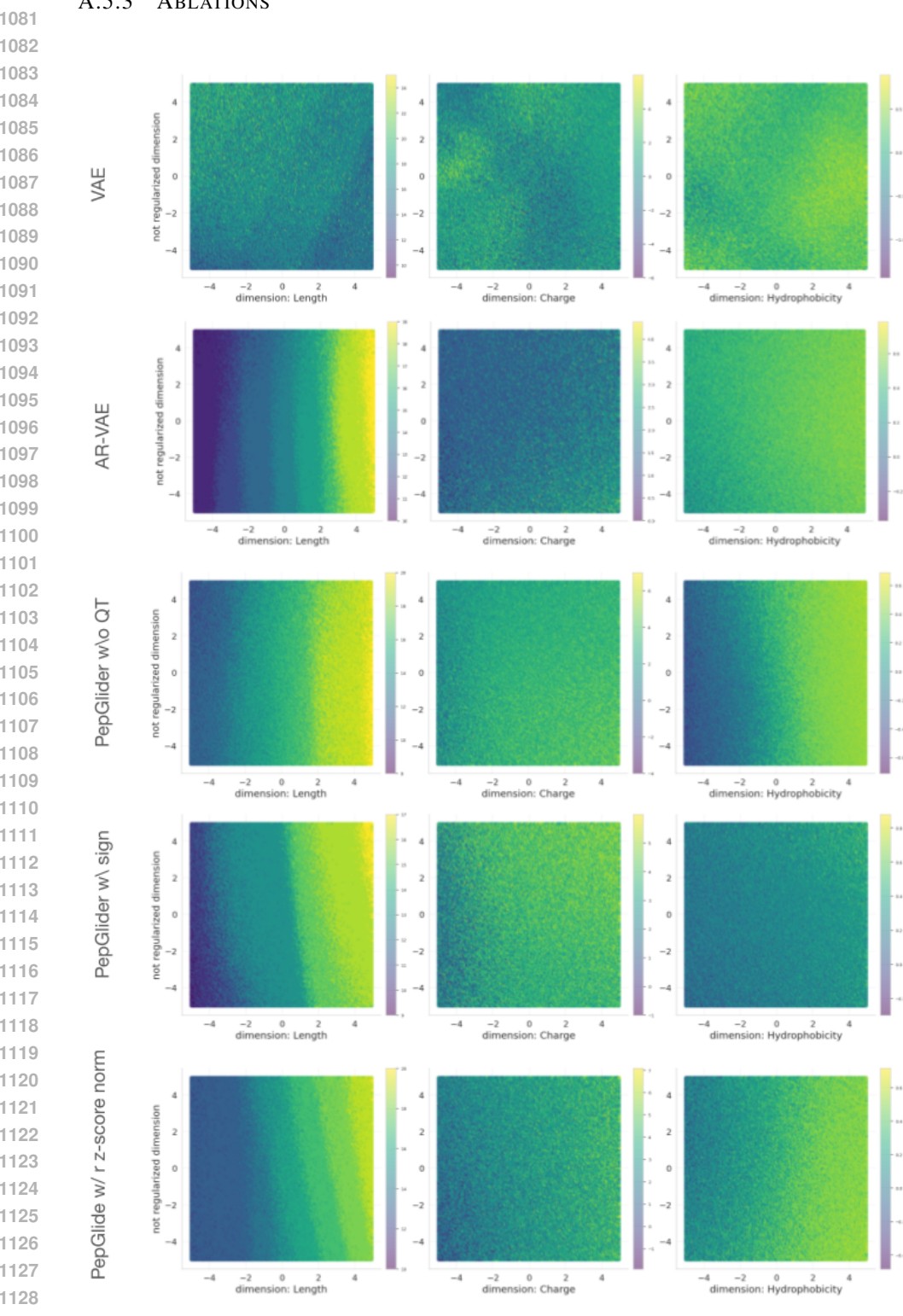

Figure 10: Latent ablations

**Latent visualizations**

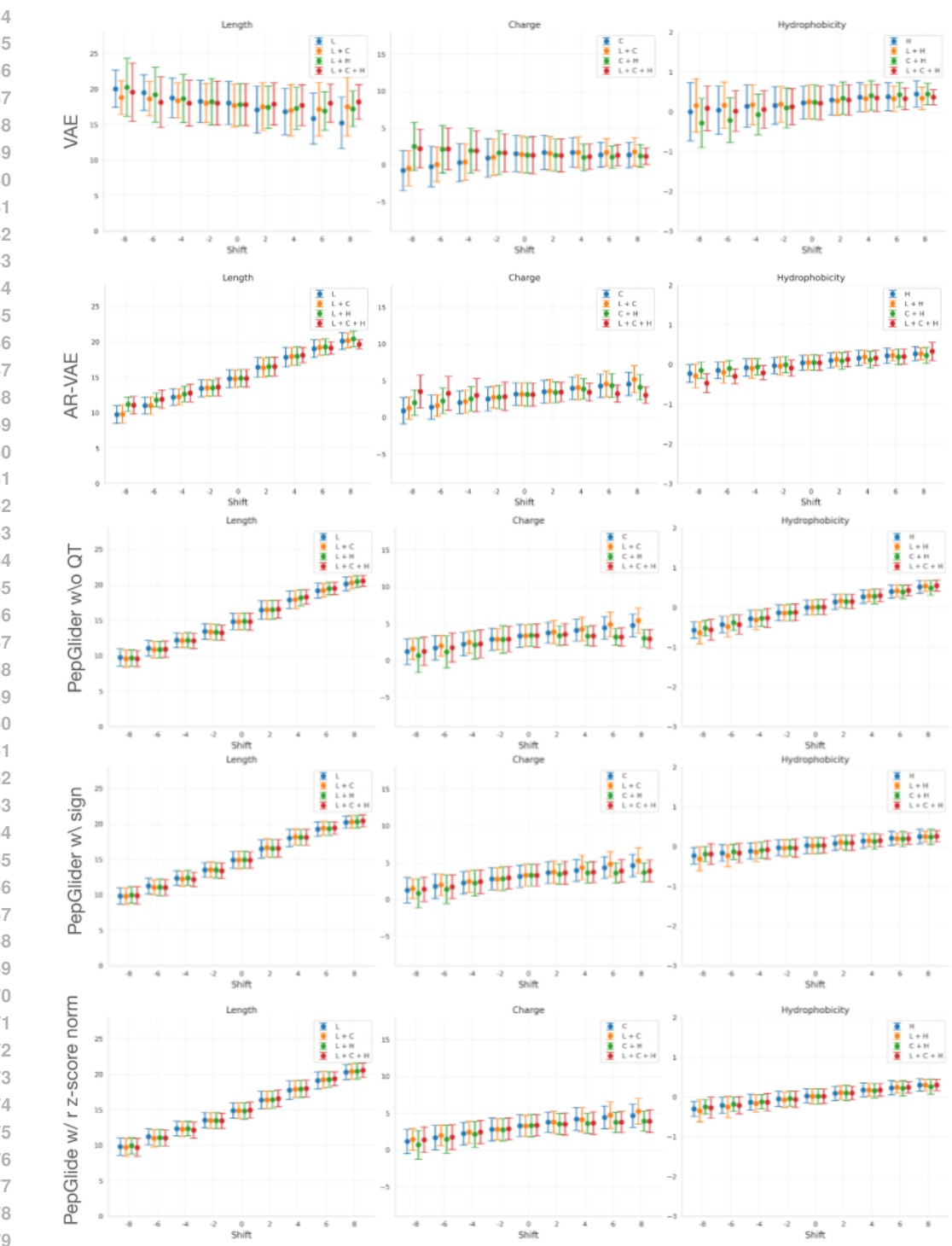

Figure 11: Multiconditioning

**Multiconditioning**

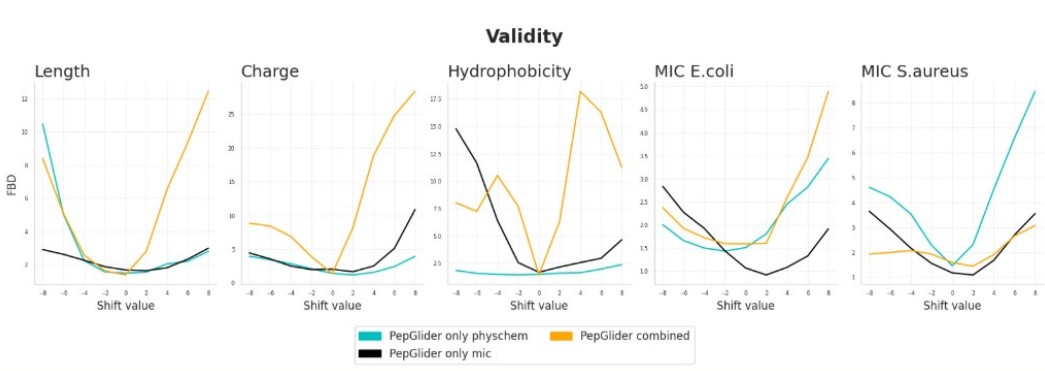

Figure 12: Caption

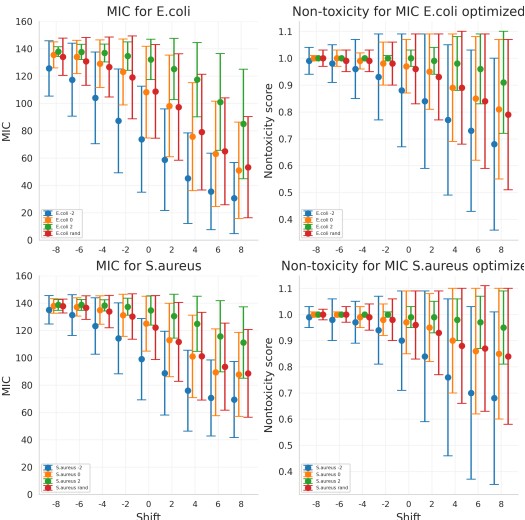

Figure 13: **Multi-objective optimization of antimicrobial activity and non-toxicity.** MIC predictions for *E. coli* (upper left) and *S. aureus* (lower left) when manipulating respective MIC-regularized dimensions, with simultaneous non-toxicity predictions (upper and lower right) under different non-toxicity regularization strategies: $\alpha = -2$ (blue), $\alpha = 0$ (orange), $\alpha = +2$ (green), and random control (red). Error bars represent standard deviation across generated peptides.

