# OpenReview forum: "PepGlider: Attribute Regularized VAE for Interpretable and Controllable Peptide Design"
_ICLR.cc/2026/Conference — ICLR 2026 Conference Withdrawn Submission_

### Official Review · Reviewer_fgWt · 2025-10-23

**Soundness:** 2
**Presentation:** 1
**Contribution:** 2
**Rating:** 4
**Confidence:** 3

**Summary:**

This paper proposes PepGlider, a peptide representation learning framework that integrates attribute regularization into deep peptide modeling for efficient binding affinity prediction.

The core idea is to jointly optimize peptide embeddings using both sequence-based supervision and attribute-guided constraints, allowing the model to learn generalizable representations transferable across peptide–protein binding datasets.

**Strengths:**

Addresses an important domain (biosequence learning).

Attribute-based regularization is intuitively reasonable and may improve generalization.

**Weaknesses:**

Attribute-based constraints are a form of similarity regularization, which has been extensively explored in metric learning and contrastive learning literature. The paper does not clearly distinguish its contribution from these prior ideas.
Without theoretical analysis, the method appears as a domain adaptation variant rather than a new learning paradigm.

The paper claims “attribute regularization preserves the manifold structure of peptide space,” but provides no mathematical justification. This weakens the scientific depth of the work.

The framework assumes that all peptides can be annotated with consistent physicochemical attributes, which is unrealistic for large-scale or novel peptide datasets.
If attributes are missing or noisy, the method may degrade sharply — yet no analysis is provided.

**Questions:**

Provide formal justification for attribute regularization (e.g., spectral regularity, cluster compactness, or contrastive geometry analysis).

Add experiments with missing attributes or noisy attributes to evaluate robustness.

---

### Official Review · Reviewer_JrA9 · 2025-10-31

**Soundness:** 3
**Presentation:** 2
**Contribution:** 2
**Rating:** 4
**Confidence:** 3

**Summary:**

This paper introduces PepGlider, a continuous attribute regularization framework that enables direct and fine-grained control over specific peptide properties. By structuring the latent space to exhibit smooth property gradients and improved disentanglement, PepGlider facilitates independent manipulation of naturally correlated attributes and supports both unconstrained generation and targeted optimization of existing peptides. Empirical results demonstrate the effectiveness of the proposed approach.

**Strengths:**

•	The three proposed technical components are theoretically grounded and carefully implemented, offering a useful methodology that could inspire future work in controllable molecular generation.

•	PepGlider achieves strong performance across multiple benchmarks, demonstrating clear advantages in controllable peptide generation compared to existing methods.

**Weaknesses:**

•	Most attribute annotations are derived from ML-based predictors. The generalization ability of these predictors on generated peptide sequences remains unclear, which may affect the reliability of the reported performance.

•	Experimental validation of antimicrobial activity is limited to two bacterial strains (E. coli and S. aureus). The rationale for selecting these specific strains is not sufficiently motivated, and it is uncertain whether they are representative enough to support broadly applicable conclusions.

**Questions:**

•	As shown in Table 2, PepGlider achieves significant improvements on sequence quality metrics in the unconstrained generation setting. Could the authors provide further analysis into the underlying reasons for these gains? Is this primarily due to the regularization strategy, the latent space structure, or other architectural factors?

**Details Of Ethics Concerns:**

While this research is aimed at designing novel antimicrobial peptides, the proposed method could potentially be misused to generate harmful or toxic biologics. The authors have appropriately acknowledged this concern in their ethics statement. It is noted, however, that such dual-use risk is common across most machine learning studies in drug discovery, and the authors have taken standard steps to address it.

---

### Official Review · Reviewer_ipvJ · 2025-10-31

**Soundness:** 2
**Presentation:** 3
**Contribution:** 2
**Rating:** 4
**Confidence:** 3

**Summary:**

This paper proposes PepGlider, which extends AR-VAE by replacing discrete sign-based attribute regularization with continuous distance alignment, combined with attribute normalization (quantile normalization and MIC-specific adaptive range normalization), aiming to achieve “precise” control of continuous peptide properties and an interpretable latent space. The authors evaluate controllability, latent space disentanglement, and multi-objective (activity/toxicity) trade-offs on antimicrobial peptides (AMPs). Experiments compare against VAE, AR-VAE, HydrAMP, Transformer-128, and demonstrate performance in both unconditional generation and “analog generation” modes.

**Strengths:**

In AMP design, controllability over continuous physicochemical properties (charge, hydrophobicity, length) and biological properties (strain-specific MIC, hemolytic toxicity) is a key challenge. Existing methods often rely on discrete conditioning or sampling-time guidance, making it hard to achieve “continuous-value targeting.” The paper focuses on this pain point and aligns well with application needs.

**Weaknesses:**

1. The shift from AR-VAE to the paper’s “continuous attribute regularization” is a relatively direct replacement of the loss together with normalization enhancements. While effective in engineering terms, there is no theoretical guarantee that this loss enables “absolute precision targeting” of property values. It still relies on mini-batch pairwise differences, lacking anchors or regression terms that map a specific latent value to a specific absolute attribute value. Without absolute calibration, “precise targeting” is hard to guarantee; what is achieved is more akin to finer-grained monotonic control.
2. Tables 1/4 show that the main model PepGlider underperforms some ablations (e.g., w/o normalization, z-score) on certain disentanglement metrics, which is not fully consistent with the claim of “improving/maintaining AR-VAE-level disentanglement.” While the authors state that AR-VAE’s strong disentanglement is preserved, claims of “significant superiority” should be made cautiously.
3. In Table 2, Diversity(↑) includes values exceeding 1 (e.g., 1.144), yet the text defines “diversity” as the proportion of unique sequences or as a distance metric—these descriptions are inconsistent (disagreement between A.4.7 and table header/notes). Novelty(↑) being 1.0 across the board is also puzzling. Definitions and computations need clarification.
4. Multiple captions/text elements contain evident errors or garbled strings (e.g., “PepLeider no Bonn,” incomplete “Caption”), which hurts readability and credibility.
5. “Code released post-publication” hinders reproducibility during review, many conclusions also depend on a proprietary validation set. While understandable, there is no publicly reproducible alternative evaluation path.
6. Beyond AMP-specific baselines, there is a lack of comparison to more general recent methods for “continuous controllable generation” or “property-controllable VAE/flow/diffusion” (e.g., approaches with explicit attribute regression heads, mutual-information-based constraints, or equivariance modeling), which weakens the evidence for methodological superiority.

**Questions:**

1. Provide more systematic Pareto analyses and quantitative metrics for multi-objective optimization (e.g., MIC distributions under a fixed toxicity threshold, or non-toxicity rates under a fixed activity threshold), not only trend plots.
2. Clearly define the formulas and value ranges for Validity, Novelty, Diversity, and AMP potential, and correct the inconsistencies and unreasonable values in Table 2.
3. Discuss and mitigate potential biases in FBD (e.g., use non-finetuned, general-purpose protein embeddings, or report consistency across multiple embedding spaces).

---

### Official Review · Reviewer_2CoX · 2025-11-01

**Soundness:** 3
**Presentation:** 1
**Contribution:** 3
**Rating:** 4
**Confidence:** 3

**Summary:**

PepGlider extends Attribute-Regularized VAEs with continuous attribute regularization to enable precise control over correlated peptide properties, moving beyond discrete conditioning mechanisms. The framework replaces signum-based loss functions with continuous formulations and introduces attribute-specific normalization to achieve exact property targeting. PepGlider achieves superior latent space disentanglement, enabling independent manipulation of naturally correlated physicochemical properties (charge, hydrophobicity, length) and complex biological attributes (antimicrobial activity, toxicity). Applied to antimicrobial peptide design, it outperforms existing methods in high-activity peptide generation, analog optimization, and activity-safety trade-offs.

**Strengths:**

1. PepGlider proposes a simple but effective modification on the existing attribute-based regularization: replacing the signum-based comparison with a continuous regularization formulation, and normalizing the attribute distance matrix to [-1,1].  It shows better continuous control. When optimizing naturally correlated properties, it can have comparable performance with the individual control on each property.
2. Comprehensive experiments verify the effectiveness of PepGlider in multiple aspects, including the disentanglement quality, continuous attribute control, multi-property control, and antimicrobial activity optimization.  Results show that compared with the baselines, PepGlider shows better disentanglement quality in all five aspects, demonstrating a stronger latent space organization and disentanglement capability for controllable generation in VAE.
3. Further studies on unconstrained generation (de novo design) and  AMP optimization show it can effectively shift the distribution towards a small MIC.

**Weaknesses:**

1. It is not clear whether the baseline methods use the same training data. For a fair comparison, the same training corpus (or at least the same scale of training data) should be used to ensure the improvement is brought by the proposed method, instead of more training data.
2. The ablation study is only conducted on the Disentanglement quality and continuous attribute control. In Table 1, the ablation variant (PepGlider w/o normalization) even shows better performance. To ensure the effectiveness of AMP designs, a similar ablation study should also be conducted.
3. Experimental results
	1. Figure 3. Lines 287-288 say that due to the inherent constraints of C + H, the control is less effective. However, in Figure 3 (c) we can also see that the non-conflicting L+H also shows inconsistent performance with the individual control of H, with a large variance.
4. A lot of revision should be done for Figures and Tables
	1. Figure 1 and a lot of the figures in the appendix are vague. More description should be added to the caption.
	2. Table 2 is small and can be enlarged.
	3. Lower panel of Figure 2: These lines are hard to distinguish from each other. Thicker lines with different markers can help the visualization. The meaning of the y-axis is unclear: higher is better or lower is better?
	4. Figure 3. The description in the caption is not consistent with the figure. The figure order is length, charge, and hydrophobicity, while the caption says charge, length, and hydrophobicity.
5. Some important related work is missing
	1. PepVAE: Variational Autoencoder Framework for Antimicrobial Peptide Generation and Activity Prediction
	2. PepCVAE: Semi-Supervised Targeted Design of Antimicrobial Peptide Sequences
	3. Artificial intelligence using a latent diffusion model enables the generation of diverse and potent antimicrobial peptides.
	4. Discovery of antimicrobial peptides with notable antibacterial potency by an llm-based foundation model.

**Questions:**

It is not clear to me why the proposed continuous regularization and the attribute normalization can also contribute to the multi-objective control. Could you provide some insights on this part?

---

### Note · Authors · 2025-11-21

I have read and agree with the venue's withdrawal policy on behalf of myself and my co-authors.